# South African University Staff and Students’ Perspectives, Preferences, and Drivers of Hesitancy Regarding COVID-19 Vaccines: A Multi-Methods Study

**DOI:** 10.3390/vaccines10081250

**Published:** 2022-08-04

**Authors:** Gavin George, Michael Strauss, Emma Lansdell, Nisha Nadesan-Reddy, Nomfundo Moroe, Tarylee Reddy, Ingrid Eshun-Wilsonova, Mosa Moshabela

**Affiliations:** 1Health Economics and HIV and AIDS Research Division (HEARD), University of KwaZulu-Natal, Durban 4041, South Africa; 2Division of Social Medicine and Global Health, Lund University, 223 63 Lund, Sweden; 3School of Nursing and Public Health, University of KwaZulu-Natal, Durban 4041, South Africa; 4School of Human and Community Engagement, University of the Witwatersrand, Johannesburg 2000, South Africa; 5Biostatistics Research Unit, South African Medical Research Council (SAMRC), Durban 4041, South Africa; 6Division of Infectious Diseases, School of Medicine, Washington University at St Louis, St Louis, MO 63110, USA

**Keywords:** vaccine hesitancy, youth, South Africa, discrete choice experiment

## Abstract

COVID-19 vaccine hesitancy poses a threat to the success of vaccination programmes currently being implemented. Concerns regarding vaccine effectiveness and vaccine-related adverse events are potential barriers to vaccination; however, it remains unclear whether tailored messaging and vaccination programmes can influence uptake. Understanding the preferences of key groups, including students, could guide the implementation of youth-targeted COVID-19 vaccination programmes, ensuring optimal uptake. This study examined university staff and students’ perspectives, preferences, and drivers of hesitancy regarding COVID-19 vaccines. A multi-methods approach was used—an online convenience sample survey and discrete choice experiment (DCE)—targeting staff and students at the University of KwaZulu-Natal, South Africa. The survey and DCE were available for staff and students, and data were collected from 18 November to 24 December 2021. The survey captured demographic characteristics as well as attitudes and perspectives of COVID-19 and available vaccines using modified Likert rating questions adapted from previously used tools. The DCE was embedded within the survey tool and varied critical COVID-19 vaccine programme characteristics to calculate relative utilities (preferences) and determine trade-offs. A total of 1836 staff and students participated in the study (541 staff, 1262 students, 33 undisclosed). A total of 1145 (62%) respondents reported that they had been vaccinated against COVID-19. Vaccination against COVID-19 was less prevalent among students compared with staff (79% of staff vs. 57% of students). The vaccine’s effectiveness (22%), and its safety (21%), ranked as the two dominant reasons for not getting vaccinated. These concerns were also evident from the DCE, with staff and students being significantly influenced by vaccine effectiveness, with participants preferring highly effective vaccines (90% effective) as compared with those listed as being 70% or 50% effective (β = −3.72, 95% CI = −4.39 to −3.04); this characteristic had the strongest effect on preferences of any attribute. The frequency of vaccination doses was also found to have a significant effect on preferences with participants deriving less utility from choice alternatives requiring two initial vaccine doses compared with one dose (β = −1.00, 95% CI = −1.42 to −0.58) or annual boosters compared with none (β = −2.35, 95% CI = −2.85 to −1.86). Notably, an incentive of ZAR 350 (USD 23.28) did have a positive utility (β = 1.14, 95% CI = 0.76 to 1.53) as compared with no incentive. Given the slow take-up of vaccination among youth in South Africa, this study offers valuable insights into the factors that drive hesitancy among this population. Concerns have been raised around the safety and effectiveness of vaccines, although there remains a predilection for efficient services. Respondents were not enthusiastic about the prospect of having to take boosters, and this has played out in the roll-out data. Financial incentives may increase both the uptake of the initial dose of vaccines and see a more favourable response to subsequent boosters. Universities should consider tailored messaging regarding vaccine effectiveness and facilitate access to vaccines, to align services with the stated preferences of staff and students.

## 1. Introduction

More than 181 million infections of SARS-CoV-2 and nearly 4 million deaths from COVID-19 have been reported as of June 2021 [1]. Although the majority of the risk for serious illness and death following infection with COVID-19 rests with older populations and populations with particular co-morbidities, younger people remain highly susceptible to infection and onward transmission [2]. The containment of COVID-19 outbreaks amongst university students has proved particularly challenging, despite instituting a number of strategies, including temporary closure and the adoption of online teaching [3]. Although limited, research has revealed that universities, in attempting to reopen, have experienced a wave of infections, with universities and the surrounding communities declared COVID-19 hotspots [4]. The availability of vaccines is viewed as an effective tool with which to control infection and reduce risk, emphasized in May 2020 by the World Health Assembly. Vaccination is viewed as an important measure to control the COVID-19 global pandemic [5], with research suggesting herd immunity through vaccination to be the most viable route to achieve epidemic control [6]. However, vaccine hesitancy poses a threat to the effective global roll-out and uptake of COVID-19 vaccines.

The vast majority of research on the topic of COVID-19 vaccine hesitancy among youth, and university staff and students, has been conducted in high-income countries, with very little known and understood about student perceptions, attitudes, acceptance, and hesitancy of vaccines in Africa [7,8,9,10,11,12]. As COVID-19 vaccines are being rolled out globally, there is a need to investigate the determinates of vaccine hesitancy, as well as vaccine service preferences among youth in different contexts.

As part of a national vaccination strategy to decentralise vaccination sites and make vaccines more accessible, the University of KwaZulu-Natal (UKZN) in South Africa was approved as a vaccination site in 2021, with a vaccination campaign launched on the 5th–6th August targeting staff members and/or students aged 35 years or older, aligned with the national roll-out strategy, who wished to be vaccinated [13]. The initial two days of implementation saw a total of 146 individuals getting vaccinated with either the Johnson & Johnson (J&J) or Pfizer vaccines [13]. Following the adjusted national COVID-19 vaccine roll-out strategy, eligibility extended to staff and students aged 18 years and above from the 30th of August 2021 [13]. A further 2340 staff and students were vaccinated, representing only a small proportion of the staff and student body [13]. Although the majority of students could not access the various university campuses, a total of 17,960 students were residing in university residences over this period. The goal was to vaccinate at least 200 staff/students per day, with the capacity of vaccinating up to 1000 staff/students per day. However, the busiest day saw only 151 vaccinations being administered, with only 16 vaccinations undertaken on the poorest performing day [13].

South African studies have revealed multiple reasons for COVID-19 vaccine hesitancy and poor vaccine uptake [14,15]. These reasons included vaccines being too new (34%), concerns about possible side effects (21%), and transparency from the government regarding the safety and effectiveness of the vaccines (14%). The latter was exacerbated by the suspension of the initial roll-out of the AstraZeneca vaccine over its effectiveness, and the J&J vaccine being put on hold temporarily due to concerns about rare clotting disorders [14].

Although these vaccination strategies may be sufficient for those who already intend to be vaccinated, for those who remain sceptical and uncertain, it is unclear which specific features of vaccination programmes could improve uptake rates, what inducements (such as mandates or incentives) would further encourage vaccination, or what inherent features of vaccines (including origin, dosage, or efficacy) would influence decisions to vaccinate.

To explore which vaccine characteristics and implementation features are most important to UKZN staff and students, particularly those who are vaccine-hesitant, we conducted a survey and discrete choice experiment (DCE) to identify the relative importance of vaccine characteristics and implementation strategies to determine how best to design vaccination programmes aligned with staff and student preferences to ensure maximal uptake, lessons which could potentially be applied to higher education institutions throughout South Africa.

## 2. Materials and Methods

### 2.1. Study Design and Setting

A multi-methods approach was carried out, involving an online survey and a DCE, to determine the prevalence, attitudes, perspectives, and preferences of COVID-19 and available vaccines among staff and students at UKZN, South Africa. The staff and student complements at UKZN in 2021 were 4403 and 44,313 respectively [16]. Potential respondents were sent a link to the online questionnaire via email through the internal notice system, and the data were collected from 18 November to 24 December 2021.

The online questionnaire was developed following a review of similar studies that evaluated students’ perspectives of, and attitude towards COVID-19 vaccines [17]. A DCE is a quantitative technique from the field of behavioural economics, used to elicit participant preferences for goods or services [18,19,20]. DCEs use stated preference data from a series of hypothetical choices made by individuals in which they make trade-offs between the different characteristics (or attributes) of a good or service—in this case, the service delivery model characteristics and inherent features of COVID-19 vaccines and services. Lancaster’s Theory of Consumer Choice [21,22] and Random Utility Theory [23] are the theoretical frameworks that underpin the design and analysis of DCEs, which assume that participants will make choices that maximise the benefit (or utility) they gain from using a good or service, and where the total utility an individual gains when making a choice comprises the utility associated with each of the attributes of the good or service on offer.

The attributes included in this study were initially drawn from a review of previous COVID-19 DCEs and refined through a series of meetings and discussions with the study team and key stakeholders at UKZN [24,25,26,27,28,29]. The final list of attributes and the possible values each attribute could take (the levels) is illustrated in Table 1.

A fractional factorial design was developed with 32 choice sets using the dcreate command in STATA, which maximises the D-efficiency of the statistical design using the modified Fedorov algorithm [30,31,32]. A blocking variable was included in the design to divide the 32 choice sets into four versions so that each participant made eight choices between two hypothetical COVID-19 vaccine models illustrated on a “choice card” (an example of a choice set is presented in the Appendix A). Each scenario offered a forced choice (no opt-out), and once participants had made their decision, they were asked whether they would really get vaccinated if offered the alternative they had just selected.

### 2.2. Sample Size

Only participants who had not already been vaccinated were eligible to answer the DCE. Thus, the sample size for the DCE depended on the total number of staff and students who were willing to take the survey and the vaccination uptake rate in that sample. A rule of thumb commonly used to determine statistical power for DCEs suggests that the minimum number of participants per stratification *n* should be at least $500*\frac{L}{S*J}$, where *L* is the maximum number of levels for any of the alternatives, *S* is the number of alternatives in any choice set, and *J* is the number of choice sets shown to each participant [33].

### 2.3. Recruitment and Data Collection

The online survey and DCE were administered to a convenience sample of staff and students at UKZN, a large South African public university. Participants were recruited through the University’s social media pages and the university’s internal notice system. Only current staff members and students enrolled at the university at the time of the study were invited to participate. Survey participants could download a letter of information which included details of the study objectives, data protection, and confidentiality. Participants were offered compensation in the form of entry into a draw for one of ten ZAR 500 (~USD 33) shopping vouchers; the draw was not linked to participants’ survey responses. Consent was indicated by participants reading the consent form on the opening page of the survey and clicking “Agree”, which enabled them to begin answering the survey questions.

### 2.4. Data Analysis

All variables were treated as categorical variables. There were two levels of gender identification (male and female), and two levels of UKZN association (staff and student). The level of intention to get vaccinated, for participants who identified as unvaccinated, was grouped as: definitely not (1), probably not (2), probably (3), and definitely (4). COVID-19 vaccine attitudes and COVID-19 perspectives were both grouped as: strongly disagree (1), disagree (2), indifferent (3), agree (4), and strongly agree (5). COVID-19 vaccine attitudes and COVID-19 perspectives had a Cronbach’s alpha of 0.9 and 0.7, respectively. Thus, both scales demonstrated an acceptable internal consistency (Cronbach’s alpha > 0.6) [34]. Data analysis was undertaken using STATA version 17 (Texas, United States of America) and *p*-values of <0.05 were considered statistically significant. To determine if there was a significant difference between the average responses from two groups (staff and students), *t*-tests were conducted. The null hypothesis being the difference was equal to zero. The alternative hypothesis being the difference was not equal to zero. Those who did not give information about whether they were staff or a student were excluded from the analysis.

The DCE analysis was conducted using a mixed effects logit model to estimate the relative utility for each attribute and level, where all attribute level coefficients were modelled as random parameters and 1000 Halton draws were used for the simulation. Attribute levels were dummy-coded and mean utilities were estimated relative to an arbitrarily assigned reference level for each attribute (shown in Table 1). Standard deviation estimates generated by the mixed effects logit model show the magnitude and significance of any heterogeneity in mean preferences within the sample and give an indication of where further analysis may be needed to identify sources of heterogeneity.

## 3. Results

Respondent characteristics were disaggregated by university staff and students (see Table 2). In total, 1836 respondents completed the survey, the majority (69%) of whom were students. Table 3 presents data on the number of staff and students who had tested positive for COVID-19, with reported rates being relatively low, with only 16% of staff and 12% of students confirming that they had received a positive COVID-19 diagnosis, with the majority (89%, data not shown) experiencing self-reported mild symptoms. The majority of staff (79%) and students (57%) had been vaccinated. Of the 555 unvaccinated respondents, 20.72% declared they would definitely vaccinate, 32.43% stated they would probably vaccinate, with 27.39% suggesting they would probably not, and 19.46% declaring they would definitely not vaccinate. The top five statements unvaccinated respondents selected (data not shown) for not getting vaccinated were; “I do not believe that the vaccine is effective” (22%), “I do not think the vaccine is safe” (21%), “I have heard/read negative media about the vaccine” (14%), “Religious reasons” (10%), “Someone else told me they had a bad reaction to the vaccine” (10%), and “I intend to use traditional or other health treatments” (9%). Respondents were asked to select which vaccine information sources they trusted, with the majority (36%) stating healthcare workers (HCWs), followed by scientific articles (26%). Few respondents sourced trusted information from mainstream (4%) and social media (0.5%), respectively. Of concern, 15% of respondents stated that they did not trust any source of information.

Table 4 shows staff and student perspectives towards COVID-19 and measures adopted to mitigate infection. Staff agreed significantly more (M = 2.29, SD = 0.96) than students (M = 2.14, SD = 0.97) with the statement that they would die if they were infected with COVID-19, *t* (1312) = 2.6421, *p* = 0.008, *r* = 0.08. Students were significantly more satisfied (M = 3.84, SD = 1.04) with the measures implemented by UKZN to prevent the spread of COVID-19, *t* (1534) = −3.1409, *p* = 0.001, *r* = 0.15, than staff (M = 3.66, SD = 1.08). Although the majority of both staff and students strongly agreed that they could easily wear a mask when in public, students were more confident (M = 4.40, SD = 0.97), *t* (1534) = −2.5610, *p* = 0.010, *r* = −0.14, than staff (M = 4.52, SD = 0.82).

Table 5 reveals staff and student attitudes towards vaccines and vaccine mandates. While both the majority of staff and students strongly agreed that getting vaccinated was effective and potentially lifesaving; the greater proportion of staff agreed that vaccines saved lives, *t* (1508) = 3.7515, *p* = 0.000, *r* = 0.20 and that vaccines were effective, *t* (1507) = 5.2033, *p* = 0.000, *r* = 0.28. Staff agreed significantly more (M = 3.70, SD = 1.11) than students (M = 3.51, SD = 1.07) with the statement that pharmaceutical companies had developed safe and effective COVID-19 vaccines, *t* (1507) = 3.1246, *p* = 0.000, *r* = 0.17. Students (M = 2.85, SD = 1.04) agreed significantly more than staff (M = 2.67, SD = 1.09) with the statement that COVID-19 vaccines carried more risks than other vaccines, *t* (1507) = −3.0654, *p* = 0.002, *r* = −0.17. Staff agreed significantly more (M = 4.05, SD = 1.18) than students (M = 3.84, SD = 1.18) with the statement that getting vaccinated was a good way to protect their family from COVID-19, *t* (1507) = 3.2190, *p* = 0.001, *r* = −0.17. Students, however, were significantly more concerned (M = 3.69, SD = 1.13) than staff (M = 3.14, SD = 1.26) about serious adverse effects of the COVID-19 vaccine, *t* (1507) = −8.4494, *p* = 0.000, *r* = −0.46. Although the majority of both staff and students were against mandatory vaccination, students agreed significantly more (M = 3.91, SD = 1.28) than staff (M = 3.02, SD = 1.45) that people should not be forced to get vaccinated, *t* (1507) = −11.9378, *p* = 0.000, *r* = −0.66.

Results from the DCE are illustrated in Figure 1, which shows the main effects results of the mixed effects logit model.

Figure 1 shows that, on average, staff and students’ preference structures are significantly influenced by vaccine effectiveness—participants were far less likely to choose alternatives with vaccines that were partially effective (at 50%) (β = −3.72, 95% CI = −4.39 to −3.04) compared with vaccines that were very effective (90%), and this characteristic had the strongest effect on preferences of any attribute. Participants also derived less utility from alternatives that were moderately effective (70%) (β = −1.38, 95% CI = −1.84 to −0.91) compared with alternatives that were very effective (90%). The frequency of boosters was found to have a significant effect on preferences, with participants deriving less utility from choice alternatives that required annual boosters (β = −2.35, 95% CI = −2.85 to −1.86) or boosters every five years (β = −1.22, 95% CI = −1.66 to −0.79) compared with alternatives that only required one-off vaccination. The influence of the number of boosters on preferences was stronger than the influence of the number of doses; however, this was still significant, with participants avoiding alternatives that required two doses (β = −1.00, 95% CI = −1.42 to −0.58) compared with alternatives that required only one dose.

Although a small incentive of ZAR 50 (USD 3.33) did not have a significant effect on preferences on average, a more generous incentive of ZAR 350 (USD 23.28) was found to have a positive effect, with participants deriving more utility from alternatives offering a ZAR 350 incentive (β = 1.14, 95% CI = 0.76 to 1.53) compared with alternatives offering no incentive. As expected, participants preferred shorter waiting times, deriving less utility from alternatives that included a five hour waiting time (β = −1.33, 95% CI = −1.77 to −0.89) or a three hour waiting time (β = −1.30, 95% CI = −1.73 to −0.86) compared with alternatives with just a one hour waiting time. However, the difference in utility between a three hour waiting time and five hour waiting time was not found to be significant. On average, the location of vaccination and vaccine origin were not found to have a significant influence on preferences.

## 4. Discussion

This study was undertaken within the context of the slow take-up of vaccination in South Africa, with only 45% of the adult population, and of concern, only 31% between the ages of 18 and 23 having been vaccinated as of the 13th of May 2022 [35]. This study aimed to provide insights into this slow uptake, with a focus on understanding university staff and student perceptions of COVID-19, along with their attitudes towards and preferences of COVID-19 vaccines. To fulfil that intent, we adopted a multi-methods approach, utilizing both a survey and DCE to determine the drivers of hesitancy and potential avenues to improve vaccination rates, specifically among young adults. Encouragingly, the majority of university staff and students had been vaccinated; however, aligned with national and international trends [36,37], vaccination rates were still lowest among students and individuals under 35 years of age. This could be linked to the supply of vaccines in South Africa, as has been the case globally, adopting a staggered approach to vaccine roll-out, initially targeting the older age cohorts and exposed groups, including health workers. At the time of the study, those aged 18 and older had only been in a position to access vaccines for three months. Notwithstanding, uptake levels were poor.

The dominant concerns amongst unvaccinated respondents in this study centred on the vaccine’s safety and effectiveness. These findings mirror other studies that have explored vaccine hesitancy among students [9,10], and children, adolescents, and young adults more broadly [37], but offers additional and contextually relevant insights regarding HCWs as trusted messengers, incentives, the appetite for mandates, and the preference architecture, all of which could guide future vaccination strategies.

Although the majority of respondents remained concerned about contracting COVID-19, only a small proportion were fatalistic about potential infection. Individual’s self-efficacy, reflected by their ability to adhere to public health measures, specifically mask wearing and physical distancing, was encouragingly high. Although the risk of severe disease and death from COVID-19 is lower in young people [35], high infection rates and low vaccination rates, together with the increased propensity to socialise, means that this cohort remains a significant vector for transmission [4]. Communication and messaging, in particular, are key to offset optimism bias, with the need to focus on vaccine safety and effectiveness, and the wider societal benefits of vaccination in protecting older family members, and vulnerable friends and colleagues [36]. Gain framed messaging has proved more effective elsewhere when espousing the benefits of vaccination, including the freedom to attend sporting events, access to entertainment venues, as well as the ability to travel are reinforced [38]. This research revealed that HCWs were the most trusted source of information, a finding similar to that found in other studies [39,40]. Understanding how to frame messages to maximise effect, whilst knowing which sources are considered reliable, remains crucial in designing communication strategies aimed at young adults.

This study further provides useful insights into which vaccination programme characteristics potentially factor into the decision to vaccinate and how these can be harnessed to inform vaccine delivery strategies. The results of the DCE showed few trade-offs with certain vaccine characteristics (effectiveness, incentives, dosage, and the need for boosters), influencing choices. These data are congruent with the survey data, highlighting the importance of the vaccine’s effectiveness to respondents. The preference for single-dose vaccines without the requirement for further boosters is further affirmed by South Africa’s vaccine statistics, which indicate a waning enthusiasm for booster shots [35]. Our evidence therefore suggests that single-dose vaccines, along with efficient vaccination services and moderate value incentives (ZAR 350; USD 23.28) could potentially improve vaccination uptake. Financial incentives have proven effective in encouraging the uptake of vaccines elsewhere [41,42,43,44]. Their feasibility and acceptability, however, are yet to be determined in the South African context. Future research on the minimum incentive feasibility and acceptability for financial and non-financial incentives is warranted. Unvaccinated staff and students remained indifferent to the location of the service and origin of the vaccine.

The well-being of university staff and students remains paramount, with the recognition that the disruption of education could result in broader social and economic consequences [45]. Many staff and students remain immunologically naïve, raising the spectre of continued temporary suspension of contact teaching following continued outbreaks. There remain strong advocates for vaccine mandates in universities [46]; however, this would need to be carefully implemented, taking into consideration who institutes the mandate and the consultation process, how to support vaccination, how to deal with exemptions and non-compliance, and how to evaluate impact which would inform the continuance of the mandate [47]. Both staff and students were largely unsupportive of vaccine mandates, revealing the current lack of buy-in for such a strategy.

Taken together, these data offer insights into who the trusted messengers are and what information, along with which vaccination service characteristics, could influence uptake. Communication targeting students will need to emphasise the value of getting vaccinated, and the subsequent need for boosters, whilst elucidating the effectiveness of vaccines and quelling fears over the vaccine’s potential side-effects. Vaccines will need to be easily accessible, and service provision efficient. Lastly, this study reveals how incentives could potentially increase vaccination rates in university settings in South Africa, particularly among students.

### Strengths and Limitations

The survey participants were a convenience sample of staff and students drawn from one university in the province of KwaZulu-Natal, and thus may not be suitable for generalizations to be drawn about other university contexts in South Africa. Participation in this study by staff and students was potentially negatively influenced by the timing of the study. The study period coincided with the examination period, with most students no longer accessing university services in December, and many staff taking leave from the middle of December. However, the sample sizes obtained were nevertheless sufficient to meet power calculations based on the key outcomes of the study.

We should also be cautious with respect to the conclusions drawn from DCEs, as they present hypothetical choices to participants, thereby limiting certainty about whether what participants think they would do would in fact translate into action. Furthermore, none of the participants in the DCE were already vaccinated; thus, their preferences were based on perception rather than experience. Although this was a limitation, participants’ perceptions were likely to ultimately drive action and underpin their decision about whether or not to vaccinate. DCEs are limited by the attributes that are included in the design. Thus, there may be other attributes that contribute to utility that were not investigated using this design (such as issues around vaccine safety). We also asked survey participants to express their attitudes and preferences towards vaccines at a time when vaccines had not long been made available, acknowledging that it is possible that both attitudes and preferences will evolve. Despite these limitations, this research provides data on an important subsection of the population and provides direct insights for local vaccination strategies. Although these limitations are important to note, the use of multi-methods helps strengthen the findings of the study. The DCE data support the findings from the survey, revealing that vaccine effectiveness is not only a concern, but plays a significant role in determining vaccine choice behaviour.

## 5. Conclusions

Understanding university students’ attitudes towards COVID-19 vaccines, together with their preferences for vaccination services, is important because vaccine uptake among this cohort remains sub-optimal. This study provides useful insights for the development of tailored messaging and services targeting young adults, with universities having the potential to play an important facilitating role in improving vaccine uptake rates among their staff and students.

## Figures and Tables

**Figure 1 vaccines-10-01250-f001:**
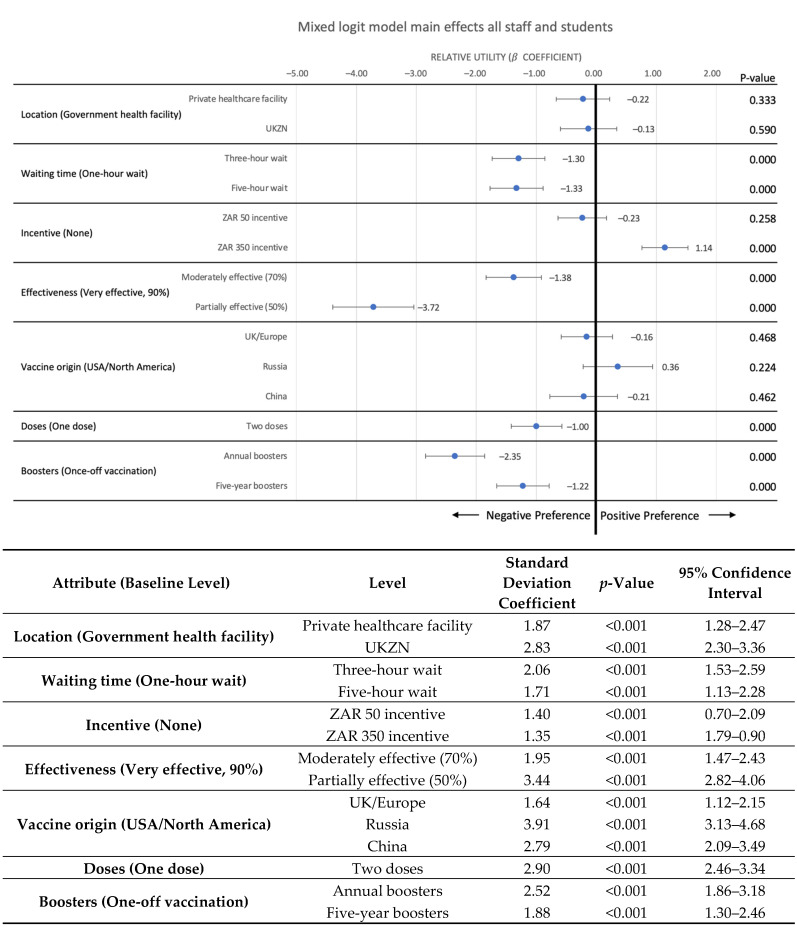
Main effects mixed effects logit model results—mean relative utility estimates and standard deviations. Notes: The point estimates show the mean relative utility (or beta coefficients) for each attribute level, and the error bars show the 95% confidence interval for each attribute level, relative to the baseline level for each attribute (shown in brackets on the left). Positive utilities represent what participants prefer, and negative utilities represent what participants do not prefer. The table on the right-hand side of the figure shows the standard deviation coefficients and *p*-values, which indicates where preference heterogeneity exists within the sample. Significant *p*-values indicate significant heterogeneity, which should be further explored to understand more clearly why preferences diverge, and which key sample sub-groups have specific preferences for particular characteristics.

**Table 1 vaccines-10-01250-t001:** Discrete choice experiment attributes and levels.

Attribute	Definition	Attribute Levels
**Vaccination location**	Location/venues where vaccine services are provided	Government–local clinic or hospital, mobile clinics
		Private hospital, family doctor, or pharmacy
		At a UKZN vaccination site
**Waiting time at vaccination site**	Length of time taken to complete the vaccination process	1 h
		3 h
		5 h
**Incentive for vaccination**	An amount provided as reward for getting vaccinated	No fee or incentive
		ZAR 50 (USD 3.33)
		ZAR 350 (USD 23.28)
**Protection against serious infection (resulting in hospitalization, ICU admission, or death)**	Percentage reduction in serious disease cases in a vaccinated group of people	Very effective (90%)
		Moderately Effective (70%)
		Partially Effective (50%)
**Vaccine origin**	Country/Region where the vaccine was developed	USA/North America
		UK/Europe
		Russia
		China
**Number of doses**	Number of vaccine shots required in order to complete the regimen	One dose
		Two doses
**Boosters required**	Frequency of additional vaccine booster shots required	One vaccination provides life-long immunity (no boosters)
		A booster required every 5 years
		Annual booster vaccinations required

ZAR 1 = USD 0.0665007: https://www.xe.com/currencyconverter/convert/?Amount=50&From=ZAR&To=USD (accessed on 23 February 2022).

**Table 2 vaccines-10-01250-t002:** Sociodemographic characteristics of respondents by UKZN association.

Characteristic Measures		Staff (*N*)	Students (*N*)	Prefer Not to Answer (*N*)
**Gender**	Male	188 (34.8%)	438 (34.7%)	13 (39.4%)
	Female	351 (64.9%)	812 (64.3%)	14 (42.4%)
	Other	1 (0.2%)	5 (0.4%)	1 (3%)
	Prefer not to answer	1 (0.2%)	7 (0.6%)	5 (15.2%)
**Age**	<35 years	106 (19.6%)	1140 (90.3%)	15 (45.5%)
	35–49 years	230 (42.5%)	104 (8.2%)	9 (27.3%)
	50 years or older	204 (37.7%)	14 (1.1%)	5 (15.2%)
	Prefer not to answer	1 (0.2%)	4 (0.3%)	4 (12.1%)
**Race**	African	215 (39.7%)	969 (76.8%)	16 (48.5%)
	Coloured	26 (4.8%)	25 (1.9%)	0
	Indian	131 (24.2%)	210 (16.6%)	3 (9.1%)
	White	137 (25.3%)	28 (2.2%)	7 (21.2%)
	Other	6 (1.1%)	9 (0.7%)	1 (3%)
	Prefer not to answer	26 (4.8%)	21 (1.7%)	6 (18.2%)
**Nationality**	South African	495 (91.5%)	1,176 (93.2%)	24 (72.7%)
	Non-South African	44 (8.1%)	78 (6.2%)	3 (9.1%)
	Prefer not to answer	2 (0.4%)	8 (0.6%)	6 (18.2%)

**Table 3 vaccines-10-01250-t003:** Prior COVID-19 testing and vaccination among respondents.

Characteristic Measures		Have You Ever Tested Positive for COVID-19?			Have You Been Vaccinated for COVID-19?		
		Yes	No	Prefer Not to Answer	Yes	No	Prefer Not to Answer
**Staff/student**	Staff	80 (16.4%)	403 (82.8%)	4 (0.8%)	429 (79.3%)	94 (17.4%)	17 (3.1%)
	Student	126 (11.9%)	923 (86.8%)	14 (1.3%)	716 (56.7%)	455 (36.1%)	71 (5.6%)
	Prefer not to answer	1 (3.6%)	27 (96.4%)	0	18 (54.6%)	6 (18.2%)	7 (21.2%)
**Gender**	Male	69 (12.5%)	476 (86.2%)	7 (1.3%)	385 (60.3%)	208 (32.6%)	36 (5.6%)
	Female	137 (13.6%)	862 (85.4%)	11 (1.1%)	767 (65.2%)	343 (29.1%)	55 (4.7%)
	Other	0	6 (100%)	0	4 (57.1%)	2 (28.6%)	1 (14.3%)
	Prefer not to answer	1 (10%)	9 (90%)	0	7 (53.9%)	2 (15.4%)	3 (23.1%)
**Age**	<35 years	119 (11.2%)	928 (87.6%)	13 (1.2%)	708 (56.2%)	458 (36.3%)	78 (6.2%)
	35-49 years	59 (18.9%)	251 (80.2%)	3 (100%)	250 (72.9%)	77 (22.5%)	11 (3.2%)
	50 years or older	29 (14.5%)	170 (85%)	1 (0.5%)	202 (90.6%)	18 (8.1%)	3 (1.4%)
	Prefer not to answer	0	4 (80%)	1 (2%)	3 (33.3%)	2 (22.2%)	3 (33.3%)

**Table 4 vaccines-10-01250-t004:** COVID-19 perspectives.

COVID-19 Perspective		M	SD	*p*-Value	*r*
**If I get COVID-19, I could get severe symptoms**	Staff	3.28	1.14	0.157	−0.05
	Student	3.18	1.17		
**If I get severe symptoms, healthcare providers will take care of me**	Staff	3.78	0.92	0.139	−0.17
	Student	3.69	1.04		
**I am worried about getting COVID-19**	Staff	3.63	1.15	0.256	0.08
	Student	3.55	1.27		
**If I get COVID-19, I will die**	Staff	2.29	0.96	0.008 *	0.08
	Student	2.14	0.97		
**I am happy with the measures implemented by the government to prevent the spread of COVID-19**	Staff	3.20	1.16	0.286	0.06
	Student	3.27	1.19		
**I am happy with the measures implemented by UKZN to prevent the spread of COVID-19**	Staff	3.66	1.08	0.001 *	0.15
	Student	3.84	1.04		
**I can easily wear a mask when in public**	Staff	4.40	0.97	0.010 *	−0.14
	Student	4.52	0.82		
**I can easily socially distance when in public**	Staff	4.04	1.08	0.433	−0.04
	Student	4.09	1.11		

* *p* < 0.05.

**Table 5 vaccines-10-01250-t005:** COVID-19 vaccine attitudes.

COVID-19 Vaccine Attitude		Mean	SD	*p*-Value	*r*
**Getting vaccinated could save my life**	Staff	4.04	1.20	0.001 *	0.20
	Student	3.79	1.19		
**COVID-19 vaccines are effective**	Staff	3.90	1.11	0.001 *	0.28
	Student	3.58	1.10		
**Pharmaceutical companies have developed safe and effective COVID-19 vaccines**	Staff	3.70	1.11	0.001 *	0.17
	Student	3.51	1.07		
**Vaccines made in Europe or America are safer than those made in other countries**	Staff	2.67	0.97	0.237	0.06
	Student	2.60	0.97		
**COVID-19 vaccines carry more risks than other vaccines**	Staff	2.67	1.09	0.002 *	−0.17
	Student	2.85	1.04		
**The information I receive about COVID-19 vaccines from Government is reliable and trustworthy**	Staff	3.28	1.12	0.593	−0.02
	Student	3.31	1.09		
**Getting vaccinated is a good way to protect my family from COVID-19**	Staff	4.05	1.18	0.001 *	0.17
	Student	3.84	1.18		
**I would do what my doctor or health care provider recommended about the COVID-19**	Staff	4.04	1.00	0.295	0.05
	Student	3.98	1.01		
**I am concerned about serious adverse effects of the COVID-19 vaccine**	Staff	3.14	1.26	0.001 *	−0.46
	Student	3.69	1.13		
**People should not be forced to get vaccinated**	Staff	3.02	1.45	0.001 *	−0.66
	Student	3.91	1.28		

* indicates significance.

## Data Availability

The data presented in this study are available on request from the corresponding author.

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
