# Peer review of "South African University Staff and Students’ Perspectives, Preferences, and Drivers of Hesitancy Regarding COVID-19 Vaccines: A Multi-Methods Study"

_vaccines, 2022, doi:10.3390/vaccines10081250_

Round 1
Reviewer 1 Report
The paper is clear and well-written and provides an interesting insight into perspectives and attitudes about COVID-19 vaccines in a sample of South Africa university students and staff.
Introduction
The introduction is well written and acts as a good premise for the rest of the manuscript.
Methods
The methods section provides a clear explanation of the discrete choice experiment methodology. I would suggest mentioning the statistical test used for the survey analysis.
Results
The study results are expressed in a clear way. However, a table number in the text does not match the corresponding table header (table 4 is referred as table 3 in the results section). As for the table attached to figure 1, I would suggest making the caption more compact in order to improve the reading flow.
Discussion, limitations and conclusions
The discussion is sound and consistent with the results, providing an interesting overview of the possible implications on vaccine policies.
References
A general review of references is required.
Author Response
Thank you for this positive review. Two issues were identified, namely that table 4 is referred as table 3 in the results section. This has now been corrected. We have also reviewed the references and corrected those where we found errors.
Many thanks for taking the time to review this manuscript.
Reviewer 2 Report
The paper addresses and attempts to assess the problem of COVID-19 vaccine hesitancy in a university-based population. The authors are correct in their concern that such hesitancy poses a threat to the success of current vaccination programs. Vaccine effectiveness and vaccine-related adverse events are known to be potential barriers to vaccination. The paper’s aim was to determine whether tailored messaging regarding vaccination programs could influence vaccine uptake and examined University staff and students’ perspectives, preferences, and drivers of hesitancy concerning COVID-19 vaccines. The authors used an online “convenience” sample survey - multi-methods approach and discrete choice experiment (DCE) which targeted staff and students at the University of KwaZulu-Natal, South Africa. The survey and DCE were conducted in late 2021 and captured demographic characteristics as well as attitudes and perspectives of COVID-19 and vaccination using modified Likert rating questions and calculated relative preferences and trade-offs. A total complement of 48,716 staff and students were invited by email to participate in the survey. Only 1,826 (3.7%) participated (541 staff, 1262 students, 33 undetermined) and of these 62% reported that they had been vaccinated against COVID-19 and therefore eliminated them from participation. Vaccination against the COVID-19 virus was less prevalent among students compared to staff (79% staff and 57% students). The vaccine’s effectiveness (22%), and its safety (21%), ranked as the two dominant reasons for not getting vaccinated. These concerns were also evident from the DCE, with participants influenced significantly by vaccine effectiveness, with participants preferring highly effective vaccines (90% effective) as compared to those listed as 70% or 50% effective, and this characteristic had the strongest effect on preferences of any attribute. The frequency of vaccination was also found to have a significant effect on preferences. An incentive of R350 (US$ 23.28) had a positive effect, compared to no incentive. Slow uptake of vaccination among young adults in South Africa is a problem. The study offered some insights into the factors driving this hesitancy. The study could help universities tailor their messaging regarding vaccine effectiveness and facilitate access to vaccines.
Some strengths of the paper would relate to the consistency of the study population (91.5% Soth African, however, the paper was weakened with a survey response rate of only 3.7% after removing vaccinated responders. The authors could perhaps come up with possible reasons for such a low response rate.
Lines 126 and 127: “Lancaster’s Theory of Consumer Choice and Random Utility Theory” should be referenced (e.g. Lancaster KJ. A new approach to consumer theory. J Polit Econ, 1966;74:132-157; and McFadden D. Measuring willingness to pay for transportation improvements. T. Gärling, T. Laitila, K. Westin (Eds.), Theoretical Foundations of Travel Choice Modeling, Elsevier, Amsterdam, The Netherlands (1998); 339–64).
The paper describes and reports a well-designed online questionnaire regarding Covid-19 vaccination using university staff and students as subjects. The paper is intelligent, statistically sound, and provides valuable insights for future vaccination strategies.
A couple of typos need fixing...
Line 319: 'outbreak' is misspelled
Line 347: change 'longed' to 'long'
Author Response
Thank you for the positive review. The one substantive issue raised centred on the survey response rate of only 3.7% after removing vaccinated responders. We have responded by highlighting this issue as a weakness. Specifically we have stated:
“Participation in this study by staff and students was potentially negatively influenced by the timing of the study. The study period coincided with the examination period with most students no longer accessing university services in December, and many staff taking leave from the middle of December. However, the sample sizes obtained were nevertheless sufficient to meet power calculations based on key outcomes of the study. “
The reviewer also identified two typos which were both rectified.
The authors would like to thank the reviewer for taking the time to review the manuscript and provide us with feedback which has ultimately resulted in a stronger submission.